# Evaluation of Complete Pathological Regression after Neoadjuvant Chemotherapy in Triple-Negative Breast Cancer Patients with BRCA1 Founder Mutation Aided Bayesian A/B Testing Approach

**DOI:** 10.3390/diagnostics11071144

**Published:** 2021-06-23

**Authors:** Piotr Kedzierawski, Pawel Macek, Izabela Ciepiela, Artur Kowalik, Stanislaw Gozdz

**Affiliations:** 1Department of Oncology, Institute of Health Sciences, Collegium Medicum, Jan Kochanowski University, 25-713 Kielce, Poland; pawel.macek@onkol.kielce.pl (P.M.); stanislawgozdz1@gmail.com (S.G.); 2Radiotherapy Clinic, Holycross Cancer Centre, 25-734 Kielce, Poland; izabela.ciepiela@wp.pl; 3Department of Epidemiology and Cancer Control, Holycross Cancer Centre, 25-734 Kielce, Poland; 4Division of Medical Biology, Institute of Biology, Jan Kochanowski University, 25-406 Kielce, Poland; arturko@onkol.kielce.pl; 5Department of Molecular Diagnostics, Holycross Cancer Centre, 25-734 Kielce, Poland; 6Clinical Oncology Clinic, Holycross Cancer Centre, 25-734 Kielce, Poland

**Keywords:** breast cancer, triple-negative breast cancer, BRCA1, pathologic complete regression, Bayesian statistics

## Abstract

The aim of this study was to evaluate the probability of pathologic complete regression (pCR) by the BRCA1 gene mutation status in patients with triple-negative breast cancer (TNBC) treated with neoadjuvant chemotherapy. The study involved 143 women (mean age 55.4 ± 13.1 years) with TNBC. The BRCA1 mutation was observed in 17% of the subjects. The most commonly used (85.3%) chemotherapy regimen was four cycles of adriamycine and cyclophosphamide followed by 12 cycles of paclitaxel (4AC + 12T). The differences between clinico-pathological factors by BRCA1 status were estimated. Odds ratios and 95% confidence intervals for pCR vs. non-pCR were calculated using logistic regression. The probability distribution of pCR based on BRCA1 status was estimated using beta distributions. The presence of T3–T4 tumours, cancer in stages II and III, lymphovascular invasion, and the use of chemotherapy schedules other than 4AC + 12T significantly decreased the odds of pCR. It was established that there was a 20% chance that pCR in patients with the BRCA1 mutation was 50% or more times as frequent than in patients without the mutation. Thus, the BRCA1 mutation can be a predictive factor for pCR in patients with TNBC.

## 1. Introduction

Breast cancer is the most frequently diagnosed cancer and the second most common cause of cancer mortality in women worldwide [1]. Triple-negative breast cancer (TNBC) accounts for approximately 15–20% of all breast carcinomas and is immunohistochemically characterized by the lack of oestrogen receptor (ER), progesterone receptor (PR), and human epidermal growth factor receptor (HER2) (also defined by the lack of HER2 amplification by fluorescence in situ hybridization (FISH)). TNBC has a highly aggressive clinical course, with an earlier age of onset, greater metastatic potential, and poorer clinical outcomes [2]. Chemotherapy has become the main approach for the treatment of TNBC. Despite the poor prognosis of TNBC, studies have demonstrated that TNBC is more responsive to chemotherapy than any other molecular subtype [3,4,5].

At this time, an anthracycline- and taxane-based regimen remains the standard of care for TNBC patients. Neoadjuvant chemotherapy allows for the reduction of tumour volume of the primary tumour and the regional nodes, which can facilitate more options for surgical treatment. Preoperative systemic treatment is mostly applied, even in patients with an early stage. However, according to the NSABP B-27 trial, where preoperative or postoperative adriamycine (A) and cyclophosphamide (C) with docetaxel was compared, there was no significant difference in disease-free survival (DFS) and overall survival (OS) [6,7].

Pathologic complete regression (pCR) is defined as the absence of residual invasive cancer or preinvasive cancer on pathologic evaluation of the resected breast and regional lymph nodes after neoadjuvant therapy. Several trials showed that pCR predicts for long-term outcomes and is, therefore, a potential surrogate marker for DFS and OS. Achieving a pCR is more common in highly proliferating carcinomas like TNBC or HER2-positive tumours as compared to luminal tumours [8]. An analysis comprised of 12 randomized clinical trials of neoadjuvant chemotherapy in breast cancer, conducted by the Collaborative Trials in Neoadjuvant Breast Cancer international working group, showed longer event-free survival (EFS) and OS in patients who achieved a pCR [9].

Studies of neoadjuvant chemotherapy have consistently reported higher response rates in patients with TNBC than those with non-TNBC, and approximately 30–40% of patients with early-stage TNBC treated with the standard neoadjuvant anthracycline- and taxane-based chemotherapy regimen achieved a pCR after treatment [4,8,10]. A preferred chemotherapy regimen in the neoadjuvant or adjuvant setting has not yet been determined, but dose-dense and high-dose regimens seem to be more effective in patients with TNBC [11,12,13]. Taxanes and anthracyclines are the most effective schedules [14,15,16]. TNBC commonly harbours BRCA gene mutations that make them particularly susceptible to DNA-damaging compounds such as platinum drugs [17]. The results of recent trials investigating the role of platinum agents in TNBC are conflicting and do not reveal a statistically significant difference due to small populations and, likely, to toxicity and the higher rate of treatment discontinuation in platinum arm [18,19]. BRCA-related breast cancer is characterised by a more aggressive phenotype than sporadic breast cancer, and is more frequently high grade and triple-negative [17,18]. For oncologists, it is very important to know BRCA status because it is strongly connected with planning systemic and surgical treatment [20,21,22]. The aim of this paper is to evaluate the probability of pCR in triple-negative breast cancers patients, treated mostly with the recommended, neoadjuvant schedule AC followed by taxane chemotherapy, based on BRCA1 gene mutation status.

## 2. Materials and Methods

### 2.1. Verification of Data

Data (*n* = 2745) of women treated for breast cancer in 2015–2019 was verified. In the first step of the verification, all patients with breast cancer biological subtypes other than triple-negative breast cancer were removed from the database. Then, the data of women initially treated surgically was removed from the database as well. From the group of patients who received preoperative chemotherapy, 12 cases with an unknown mutation status of the BRCA gene were removed. Ultimately, 143 patients were qualified for the study. Detailed information on data verification is presented in Figure 1.

### 2.2. Pathologic Complete Regression Status Determination

The pathological response was assessed by pathologists after the completion of the neoadjuvant chemotherapy and surgery. Pathologic complete regression was defined as no evidence of residual invasive or preinvasive cancer in the breast, whether the removed whole breast after mastectomy or part of the breast after breast conserving surgery, and in the removed axillary lymph nodes (ypT0N0). 

### 2.3. BRCA Status Determination

#### 2.3.1. DNA Isolation

Whole blood collected in EDTA tubes was used for DNA isolation. DNA was isolated from 100 μL of whole blood within 12 h of collection using the Micro AX Blood Gravity Kit (A&A Biotechnology, Gdańsk, Poland). The DNA was eluted in a volume of 120 μL of buffer E.

#### 2.3.2. High-Resolution Melting (HRM)-Polymerase Chain Reaction (PCR) and Sanger Sequencing

We have used HRM for 10 years. Before HRM was introduced to detect the founder mutations in BRCA1, this method was validated based on control samples. In addition, several hundred samples were genotyped using both HRM and Sanger technologies. We obtained 100% concordance in mutation detection using both technologies. To confirm the presence of a mutation detected by HRM with Sanger, we selected more samples for confirmatory Sanger sequencing than could be estimated by analysing HRM curves, which protects against false negative results. Based on the above facts, screened founder and recurrent mutations in BRCA1 (5382insC- c.5266dupC (p.Gln1756Profs), c.5370C>T- c.5251C>T (p.R1751*), 300T>G- c.181T>G (p.Cys61Gly), 185delAG- c.68_69delAG (p.Glu23Valfs), and 4153delA- c.4035delA (p.Glu1346Lysfs)) were all detected with very high confidence.

PCR was performed using a mixture of 7 μL Qiagen Type-it PCR Polymerase Mix (Qiagen, Hilden, Germany), 5 μL water, 1 μL of each primer (Table 1), and 1 μL template DNA. The reaction was carried out in a Rotor Gene Q thermocycler (Qiagen). The reaction conditions were: 95 °C for 5 min; 40 cycles of 95 °C for 10 s, 67 °C for 30 s (touchdown of 1 °C/cycle for 10 cycles), and 72 °C for 20 s; and one cycle of 95 °C for 10 s and 40 °C for 20 s. After completion of the PCR reaction, the HRM melting reaction was performed on a Rotor Gene Q thermocycler (Qiagen, Hilden, Germany). The HRM range for the examined exons in the BRCA1 gene was 75–87 °C. The reaction results were analysed, relative to control samples, using the Rotor-Gene Q Series Software Version 2.2.3 (Qiagen, Hilden, Germany). The melting curve profiles of the tested samples were compared with the melting curves of the non-mutated and mutated control samples. All mutations detected using the HRM-PCR technique were confirmed using capillary sequencing. Curves diverging in shape from the control curve (WT) were verified by capillary sequencing. The Sanger sequencing reaction used PCR amplification products that were purified using 10 U of exonuclease I (EN 0582) and 1 U of phosphatase Fast-AP (EF 0651) (both from Thermo Fisher Scientific, Waltham, MA, USA). The reaction was incubated for 15 min at 37 °C, followed by 20 min at 80 °C. The sequencing reactions were performed using forward (F) and reverse (R) sequence-specific primers (Table 1) and the ABI PRISM Big Dye Terminator Kit, version 3.1 (catalogue number 4337450, Applied Biosystems; Thermo Fisher Scientific, Waltham, MA, USA), according to the manufacturer’s instructions. The sequencing results were analysed using the 3130 capillary sequencer (Applied Biosystems; Thermo Fisher Scientific). The generated sequences were compared to the reference sequence using the NCBI Blast Nucleotide program (https://blast.ncbi.nlm.nih.gov/Blast.cgi?PAGE_TYPE=BlastSearch (accessed on 7 April 2021)).

### 2.4. Statistical Analyses 

Basic statistics were presented as mean ± standard deviation or number and proportion, depending on the type of the studied variable. Statistical differences, depending on the mutation status of the BRCA1 gene, were estimated using a *t*-test (equal variance), Welch test (unequal variance), or the chi-square test (categorical variables). Equality of variance was checked with the F test. The odds ratios (ORs) and 95% confidence intervals (95% CI) for pCR vs. non-pCR, in relation to clinico-pathological factors, were calculated using logistic regression models. The confidence intervals were based on the log-likelihood ratio. *p* values of <0.05 were considered statistically significant. In Table 2, statistical significance is marked with asterisks (* *p* < 0.05; ** *p* < 0.01; and *** *p* < 0.001). The probability distribution of pCR based on the available data, depending on the mutation status of the BRCA1 gene, was estimated on the basis of the beta distributions. Functions describing the probability of each possible pCR hypothesis were obtained provided that α pCR events and β non-pCR events occurred for each analysed BRCA1 mutation status. Based on the available literature, in patients with triple-negative breast cancer, a BRCA1 mutation, and neoadjuvant chemotherapy, the pCR rate was ≈40% and ≈30% in patients without the BRCA1 gene mutation [23,24]. Therefore, two different prior beta distributions were adopted for the pCR probability according to the mutation status of the BRCA1 gene: beta distribution (3, 7) for patients without the mutation, and beta distribution (4, 6) for patients with the mutation. The mean probabilities of pCR for the non-mutated and mutated patients were 0.3 and 0.4, respectively. The means of the analysed beta distributions were calculated on the basis of the following formula:(1)mean beta distribution = α/(α+ β)

At the same time, in both cases, a wide range of alternative pCR probabilities was allowed. In order to obtain a posterior distribution, the reliability based on the available data was combined (the sum of the parameters of beta distributions) with the assumed prior distributions of pCR depending on the BRCA1 mutation status. The posterior beta distribution was calculated based on the following formula:(2)$$\mathrm{Beta}\left(\alpha_{\mathrm{posterior}},\beta_{\mathrm{posterior}}\right)=\mathrm{Beta}\left(\alpha_{\mathrm{likelihood}}+{\;\alpha }_{\mathrm{prior}},{\;\beta }_{\mathrm{likelihood}}+\beta_{\mathrm{prior}}\right)$$

The 95% credible intervals were established on the basis of the quantile function for the beta distribution, determining quantiles of the order of 0.025 and 0.975 separately for each analysed BRCA1 gene mutation status. The assessment of which mutation status of the BRCA1 gene generated a higher probability of pCR was performed using Monte Carlo simulation, a technique that, in this study, used random sampling from two beta distributions, where each sample was selected based on its probability in the distribution. As a result, samples from high-probability areas were selected more frequently. Sampling from two beta distributions was treated as one sample. Considering the influence of the number of trials on the accuracy of the result, 100,000 trials were taken. Then, it was compared from which distribution the samples were taken more frequently, and the result obtained was divided by the total number of samples. In this way, the percentage of all trials, where one of the two BRCA1 mutation statuses generated a higher probability of pCR than the other, was calculated. Using the simulation results, the number of times samples from the beta distribution of the BRCA1 gene mutation status, which generated a higher probability of pCR, were taken more often was estimated (Formula (3)):(3)probability superior=sum (superior samples>inferior samples)/number of trials

Based on the quotient of the analysed distributions, where the numerator was the beta distribution of the BRCA1 mutation status with a higher probability of pCR, the distribution of relative improvements between the analysed distributions was determined (Formula (4)):(4)distribution of the relative improvements= superior samples/inferior samples

The results of this analysis were illustrated by the empirical cumulative distribution, on the basis of which the 25th, 50th, and 75th percentiles were determined. All statistical analyses were performed in R (version 3.5.3).

## 3. Results

The study involved 143 women (mean age 55.4 ± 13.1 years) treated with neoadjuvant systemic treatment for triple-negative breast cancer between 2015 and 2019.The method to determine ER, PgR, and HER2 negativity was used according to the American Society of Clinical Oncology, College of American Pathologists guidelines: ER and PgR nuclear staining of less than 1% by immunohistochemistry (IHC) and HER2 IHC staining of 0 to 1+ or fluorescent in situ hybridization <2.0 if IHC 2+ or IHC not performed. The dominant type was NST carcinoma, observed in 89.5% of the patients, while T2 tumour stage was diagnosed in more than half of the patients, and no lymph node metastases were present in 62.9%. More than half of the tumours were grade III. Lymphovascular invasion (LVI) and perineural invasion (PNI) were observed in 21.7% and 2.8% of the patients, respectively. Pathological complete regression (pCR) was reported in 42.7% of the women. The most commonly used (85.3%) chemotherapy regimen was four cycles of adriamycine and cyclophosphamide, followed by 12 cycles of paclitaxel (4AC + 12T), in 122 women. In 21 patients, other schedules of chemotherapy were applied: epirubicin (E) or adriamycine (A) with cyclophosphamide and taxol (T) with carboplatin (C), or adriamycine or epirubicin with taxol or cyclophosphamide. Breast-conserving surgery was performed in 28.7% of the women. The BRCA1 mutation was observed in 17% (*n* = 24) of the subjects. Significant differences between patients with either BRCA1 mutation status were noted in age, histological grade, lymphovascular invasion, and surgical treatment methods (Table 2).

The odds of pCR occurrence were strongly associated with the tumour size, nodal status, staging, lymphovascular invasion, and the chemotherapy schedule used. The presence of T3–T4 tumours, cancers in stage II and III, lymphovascular invasion, and the use of chemotherapy schedules other than 4AC + 12T significantly reduced the chance of pCR (Table 3).

The remaining tested features showed no association with the pCR and they were not significant. The described situation was noted in the case of the BRCA1 gene mutation, the presence of which, although insignificantly, increased the odds of pCR by 75%. Since in the study group, the BRCA1 mutation was observed only in 18% (*n* = 24) of patients, and the presence of pCR in 13 patients, it was assumed that a small number of pCR and non-pCR events influenced the results of the presented estimates. For this reason, it was decided to change the data mining method to using the Bayesian approach.

Probability distributions of pCR based on the available data depending on the mutation status of the BRCA1 gene were estimated based on the beta distributions (Figure 2). The 95% credible intervals of the pCR of beta distributions for patients without the BRCA1 mutation and with the current mutation were 31.7–49.3% and 34.5–73.2%, respectively. In the case of both estimated ranges, their upper limits indicated a high, and in the case of BRCA1 positive, a very high probability of pCR. The probability of 73% of pCR in patients with BRCA1 mutation seemed particularly unlikely. To obtain beta posterior distributions, the credibility based on the available data was combined with the assumed prior distributions of pCR probability depending on BRCA1 mutation status (Formula (2)).
$$\begin{array}{c} {\mathrm{Beta}}_{posterior\;BRCA1\;negative}=\;\mathrm{Beta}\left(48+3.71+7\right)\\ {\mathrm{Beta}}_{posterior\;BRCA1\;positive}=\;\mathrm{Beta}\left(13+4.11+6\right) \end{array}$$

The posterior results indicated that in patients with the mutation, a higher probability of pCR was observed, but also a wider range of its possible probability. Based on beta posterior distributions, the 95% credible intervals of the pCR for patients without the BRCA1 mutation and with the mutation were 31.3–48.1% and 33.5–66.5%, respectively. Unfortunately, the pCR probability distributions of both analysed beta distributions overlapped. Therefore, it could not be unequivocally ruled out that pCR in patients without the mutation is much more likely than in patients with the mutation (Figure 2).

To dispel any doubts, the Monte Carlo simulation was used, automatically drawing 100,000 samples simultaneously from both analysed beta distributions. Following this, we estimated the percentage of all trials in which the beta distribution of possible pCR of mutated patients was more likely than in patients without the mutation. Based on all the tests performed, the distribution of possible pCRs for patients with the mutation was 86% better when compared to patients without the mutation (Formula (3)):sum(BRCA1 positive beta samples>BRCA1 negative beta samples)/100,000=0.86

In the frequency statistics, this result was equivalent to obtaining the value of *p* = 0.14. As this result was not statistically significant, it can be concluded that the pCR probability distributions in patients with and without the BRCA1 mutation did not differ significantly from each other. However, the lack of statistically significant differences meant no differences at all and did not specify how much the pCR probabilities by BRCA1 gene mutation status were different from each other. Using the results of the Monte Carlo simulation, the quotient of the pCR probability distributions was calculated depending on the mutation status of the BRCA1 gene (Formula (4)):$$\begin{array}{l} \text{distribution of the relative improvements}\\ =\mathrm{BRCA}1\;positive\;\mathrm{beta}\;\mathrm{samples}/\mathrm{BRCA}1\;negative\;\mathrm{beta}\;\mathrm{samples} \end{array}$$

The results are presented graphically in the form of empirical cumulative distribution (Figure 3).

The 25th, 50th, and 75th percentile values were 1.10, 1.26, and 1.44, respectively, indicating a higher probability of pCR in patients with the mutation. It was also estimated that there was a 20% chance that pCR in patients with the BRCA1 mutation was 50% or more times as frequent than in patients without the mutation.

## 4. Discussion

Pathologic complete regression is the crucial factor for patients receiving neoadjuvant chemotherapy. A recently published cohort study with prospective follow-up of patients treated with an anthracycline- and taxane-based neoadjuvant chemotherapy regimen demonstrated an estimated 10-year relapse-free survival of 86% for TNBC patients who achieved pCR, versus only 23% for those with significant residual disease after chemotherapy [24]. In our group, most patients were treated with taxane-based schedules with adriamycine and cyclophosphamide. In the BRCA-mutated group, statistically, most patients achieved pCR after AC followed by paclitaxel in comparison to no BRCA-mutated patients. As is shown in Table 1, time from the beginning of chemotherapy to its end was not prolonged and we did not notice high toxicity, which was connected with interruptions or prolongations of the systemic treatment.

As already discussed, triple-negative breast cancer is strongly associated with germline mutations in the BRCA1 gene and can be much more sensitive to platinum agents [25]. In our study group, platinum agents were added to schedules of standard chemotherapy, but the number of patients was limited. And in most patients, the systemic treatment with platinum agents was interrupted or terminated due to acute toxicity. Several non-randomized clinical trials showed that a single-agent platinum-based therapy was an active regimen in BRCA-mutated breast cancer patients, especially in those with TNBC [25,26,27,28]. However, available data on the benefit of adding a platinum agent in BRCA-mutated patients receiving standard neoadjuvant chemotherapy are more limited and controversial. So far, only two randomized trials reported pCR results according to germline BRCA mutational status [23,27]. Pooled results from these trials showed that the addition of carboplatin to paclitaxel followed by anthracycline plus cyclophosphamide was not associated with a significant increased pCR rate in BRCA-mutated breast cancer patients (OR 1.17, 95% CI 0.51–2.67, and *p* = 0.711), while the benefit was present in patients without BRCA mutations (OR 2.72, 95% CI 1.71–4.32, and *p* < 0.001). Several neoadjuvant clinical trials evaluated the impact of adding platinum to standard chemotherapy. Silver et al. showed a pCR rate of 22% among all TNBC patients given neoadjuvant cisplatin [29]. The GeparSixto trial demonstrated a significant improvement in pCR with carboplatin (*p* = 0.005) in patients with TNBC. However, the toxicities in the carboplatin arm caused a significantly higher rate of treatment discontinuation compared to the no carboplatin arm [30]. In the CALGB 40603 trial, a standard chemotherapy of weekly paclitaxel for 12 weeks followed by dose-dense doxorubicin and cyclophosphamide for four cycles was evaluated. The analysis revealed that the pCR rate was significantly increased with carboplatin (54%) compared with the control arm (41%). The survival data at 3 years did not demonstrate a significant improvement with carboplatin; however, the study was not powered to detect event-free survival [18]. The effect of cyclophosphamide was also supported by the GeparOcto trial, addressing the question of whether the high-dose intensity combination of epirubicin, taxane, and cyclophosphamide is equivalent to the carboplatin-containing treatment schedule of paclitaxel, liposomal doxorubicin, and carboplatin. This study suggested carboplatin and high-dose cyclophosphamide may be interchangeable, in combination with taxane and anthracycline, with similar pCR rates in both arms [31]. At this time, an anthracycline and taxane-based regimen remains the standard of care for TNBC patients [32,33].

This study has limitations that should be mentioned. Firstly, we decided to analyse only the impact of the presence of founder mutations in BRCA1 on pCR because they account for about 70% of all mutations detected in BRCA1 and BRCA2 genes in the Polish population [34] and mutations in BRCA1 are more frequent in TNBC [35]. Secondly, the relative deficiency of patients with the BRCA1 mutation (17%; *n* = 24) could have influenced the bias of the obtained results. Thirdly, the high percentage of pCR (54%) in the small group of patients with the current BRCA1 mutation seemed to be rather a feature of the studied group of patients compared to the population of patients with similar clinical and pathological parameters. Fourthly, the assumed prior beta distributions for pCR probability by BRCA1 mutation status were, perhaps, the most controversial aspect of this study as they are subjective. However, in this particular case, there was prior information about the problem under investigation, and the adoption of prior assumptions allowed for better inference based on a small amount of data. It was also assumed that if the adopted prior distributions were too subjective, they would be updated in the future through an iterative process of data collection.

## 5. Conclusions

As our data showed, the BRCA mutation can be the predictive factor for pCR in patients with TNBC. The chemotherapy regimen with adriamycine, cyclophosphamide, and taxans can be effectively used in triple-negative breast cancer patients, even with BRCA1 mutation. 

## Figures and Tables

**Figure 1 diagnostics-11-01144-f001:**
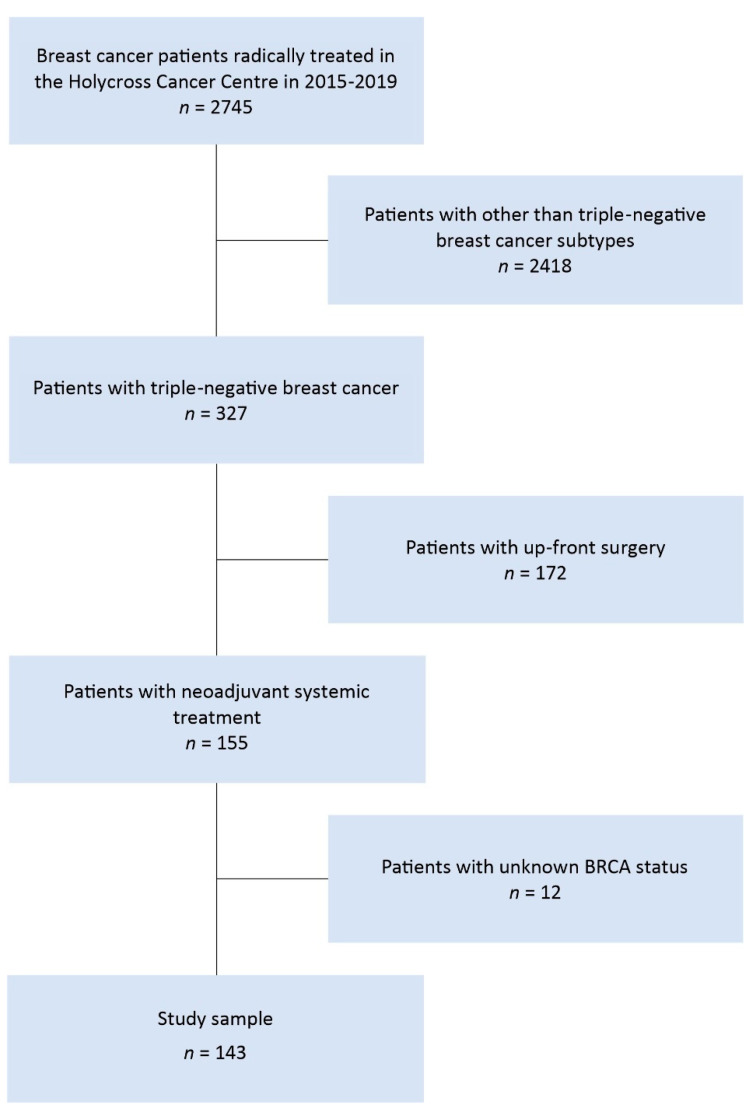
Flow diagram of data selection process.

**Figure 2 diagnostics-11-01144-f002:**
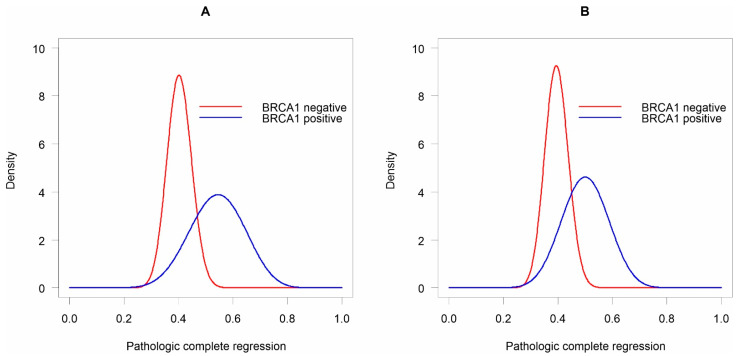
Beta distributions of pCR depending on the mutation status of the BRCA1 gene. The (**A**) available data, and (**B**) posterior distributions.

**Figure 3 diagnostics-11-01144-f003:**
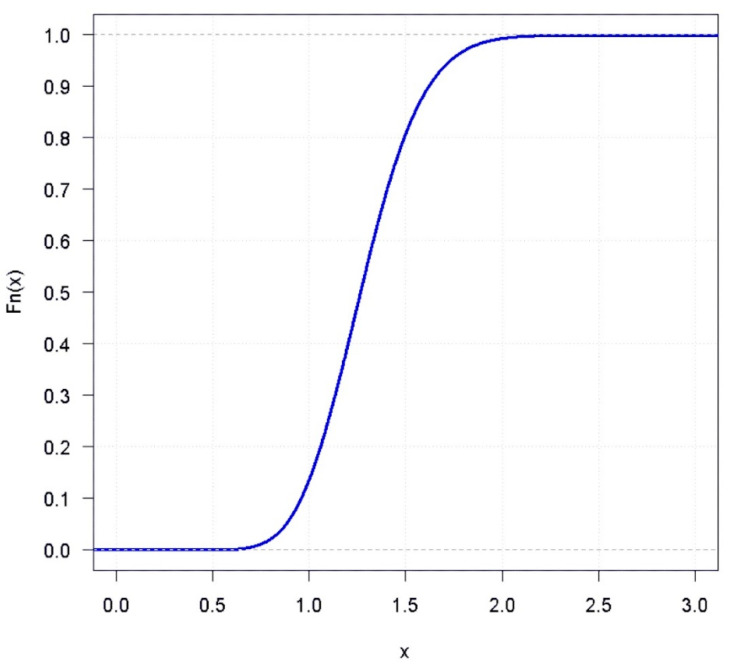
Empirical cumulative distribution.

**Table 1 diagnostics-11-01144-t001:** The primer sequences used for BRCA1 founder mutations analysis by HRM-PCR and Sanger sequencing.

Exon	Sequence-Specific Primers	Primer Sequences
*BRCA1* e20	F	ATATGACGTGTCTGCTCCAC
	R	GGGAATCCAAATTACACAGC
*BRCA1* e5	F	CTCTTAAGGGCAGTTGTGAG
	R	TTCCTACTGTGGTTGCTTCC
*BRCA1* e2	F	GAAGTTGTCATTTTATAAACCTTT
	R	TGTCTTTTCCCTAGTATGT
*BRCA1* e11	F	CAGGGAGTTGGTCTGAGTGAC
	R	GCTCCCCAAAAGCATAAAC

Abbreviations: F-forward, R-reverse, A-adenine, T-thymine, G-guanine, and C-cytosine.

**Table 2 diagnostics-11-01144-t002:** Basic characteristic of study group by BRCA1 status.

Characteristic	No BRCA1 Mutation (*n* = 119)	BRCA1 Mutation (*n* = 24)	*p*
Age (years), mean ± SD	57.1 ± 12.5	47.0 ± 12.9	<0.001
Cancer side			>0.05
Right	62 (52.1)	11 (45.8)	
Left	57 (47.9)	13 (54.2)	
Cancer type			>0.05
NST	105 (88.2)	23 (95.8)	
Other ^#^	14 (11.8)	1 (4.2)	
Ki-67, mean ± SD	64.3 ± 25.8	77.3 ± 17.6	<0.001
Tumour size			>0.05
T1	5 (4.2)	3 (12.5)	
T2	62 (52.1)	16 (66.7)	
T3	35 (29.4)	5 (20.8)	
T4	17 (14.3)	no	
Nodal status			>0.05
Nodes negative (cN−)	72 (60.5)	18 (75.0)	
Nodes positive (cN+)	47 (39.5)	6 (25.0)	
Clinical stage (CS)			<0.01
I	44 (37.0)	17 (70.8)	
II	56 (47.1)	7 (29.2)	
III	19 (16.0)	no	
Grading (G)			>0.05
G1	4 (3.4)	no	
G2	56 (47.1)	7 (29.2)	
G3	59 (49.6)	17 (70.8)	
LVI			<0.05
Yes	30 (25.2)	1 (4.2)	
No	89 (74.8)	23 (95.8)	
PNI			>0.05
Yes	4 (3.4)	no	
No	115 (96.6)	24 (100)	
CHTH			>0.05
4AC+12T	101 (84.9)	21 (87.5)	
AC/EC+ T+ Carboplatin	9 (7.6)	2 (8.3)	
Other ^†^	9 (7.6)	1 (4.2)	
Type of surgery			<0.001
BCT	40 (33.6)	1 (4.2)	
Radical mastectomy	51 (42.9)	1 (4.2)	
Simple mastectomy	7 (5.9)	3 (12.5)	
Skin sparing or nipple-sparing mastectomy	21 (17.7)	19 (79.2)	
pCR (ypT0 ypN0)			>0.05
Yes	48 (40.3)	13 (54.2)	
No	71 (59.7)	11 (45.8)	
Time 1 (days), mean ± SD	167.3 ± 73.0	161.5 ± 19.0	>0.05
Time 2 (days), mean ± SD	34.8 ± 18.0	33.0 ± 22.3	>0.05

Notes: Data is presented as number (percentage) unless stated otherwise. ^#^ category contains specific types of cancer: apocrinal, metaplastic, lobular, and medullar; ^†^ category contains: E-epirubicin, C-cyclophosphamide, F- fluorouracil, EC, FA-adriamycine, C, AC, AT- taxotere; Time 1: the time from the start to the end of chemotherapy; Time 2: the time from the end of chemotherapy to the surgery. Abbreviations: NST-no special type; LVI-lymphovascular invasion; PNI-perineural invasion; CHTH-chemotherapy; BCT-breast conserving treatment; and pCR-pathological complete regression.

**Table 3 diagnostics-11-01144-t003:** ORs (95% CIs) of pCRs vs. non-pCRs associated with clinico-pathological factors.

Characteristic	OR (95% CI)
BRCA1 mutation status	
BRCA1 positive vs. BRCA1 negative	1.75 (0.72, 4.3)
Age (years)	0.97 (0.95, 1.00)
Cancer type	
NST vs. other ^#^	1.56 (0.52, 5.23)
Ki-67	1.01 (1.00, 1.03)
Tumour size	
T2 vs. T1	0.17 (0.01, 0.99)
T3 vs. T1	0.05 (0.01, 0.31) **
T4 vs. T1	0.02 (0.01, 0.18) **
Nodal status	
Node positive vs. node negative	0.43 (0.21, 0.88) *
Staging	
Stage II vs. stage I	0.32 (0.15, 0.67) **
Stage III vs. stage I	0.12 (0.03, 0.41) **
Grading	
G2 vs. G1	0.85 (0.10, 7.47)
G3 vs. G1	0.65 (0.07, 5.67)
LVI	
Yes vs. No	0.03 (0.01, 0.14) ***
CHTH	
4AC + 12T vs. other ^†^	5.44 (1.73, 24.07) **
Time 1 (days)	1.00 (0.99, 1.00)
Time 2 (days)	0.99 (0.97, 1.01)

Notes: ^#^ category contains specific types of cancer: apocrinal, metaplastic, lobular, and medullar; ^†^ category contains different schedules of CHTH: EC, FEC, FAC, AC, AT, and schedules with carboplatin; Time 1: the time from the start to the end of chemotherapy; Time 2: the time from the end of chemotherapy to the surgery; * *p* < 0.05; ** *p* < 0.01; and *** *p* < 0.001. Abbreviations: NST-no special type; LVI- lymphovascular invasion; CHTH-chemotherapy; and BCT-breast conserving treatment.

## Data Availability

The data presented in this study are available on request from the corresponding author. The data are not publicly available due to privacy limitations.

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
