# Peer review of "Evaluation of Complete Pathological Regression after Neoadjuvant Chemotherapy in Triple-Negative Breast Cancer Patients with BRCA1 Founder Mutation Aided Bayesian A/B Testing Approach"

_diagnostics, 2021, doi:10.3390/diagnostics11071144_

Round 1
Reviewer 1 Report
This is an interesting topic that makes the manuscript attractive, and the manuscript is clearly written. The main limitation is the exclusive analysis of the BRCA1 founder mutations. It is true that triple-negative breast cancers more frequently harbor BRCA1 pathogenic variants, however, also BRCA2 PV could be found (PMID: 33403015). The same for other, non-founder mutations (PMID: 30040829). I recommend adding these 2 concepts in the final part of the discussion, among the limits of the study, citing the previous references.
Author Response
AUTHORS' RESPONSES TO THE REVIEWER'S RECOMMENDATIONS
REVIEWER # 1
Comments and Suggestions to the Authors:
This is an interesting topic that makes the manuscript attractive, and the manuscript is clearly written. The main limitation is the exclusive analysis of the BRCA1 founder mutations. It is true that triple-negative breast cancers more frequently harbor BRCA1 pathogenic variants, however, also BRCA2 PV could be found (PMID: 33403015). The same for other, non-founder mutations (PMID: 30040829). I recommend adding these 2 concepts in the final part of the discussion, among the limits of the study, citing the previous references.
AUTHORS’ RESPONSE
We are grateful reviewer for this remark.
We agree with the reviewer that the BRCA1 mutation are more frequently detected in TNBC. The aim of the study was to evaluate the probability of pCR in triple-negative breast cancers patients treated mostly with recommended, neoadjuvant schedule AC followed by taxane chemotherapy and comparison the effects in BRCA mutation negative and positive patients.
We have added the following explanation in the limitation section of the discussion.
“Firstly, we decided to analyse only the impact of the presence of founder mutations in BRCA1 on pCR because they account for about 70% of all mutations detected in BRCA1 and BRCA2 genes in the Polish population [34] and mutations in BRCA1 are more frequent in TNBC [35]”.
The spelling was checked again as suggested.
Reviewer 2 Report
I was able to read this paper with pleasure as there are not many studies that give an answer on whether the treatment outcome is different depending on the BRCA status. I have a few minor point questions, so please answer them.
1. Is clinical stage in table 2 a prechemo stage? Is it postchemo stage? If it is a prechemo stage, why was the neoadjuvant chemo performed in stage 1?
2. P6/13 spelling check –epirubicyne -> epirubicin
3. Clarfy a definition of pCR adopted in this study: No invasive tumor (+/- residual DCIS, ypT0-is) vs No carcinoma including DCIS (ypT0 only)
Author Response
AUTHORS' RESPONSES TO THE REVIEWER'S RECOMMENDATIONS
REVIEWER # 2
Comments and Suggestions to the Authors:
I was able to read this paper with pleasure as there are not many studies that give an answer on whether the treatment outcome is different depending on the BRCA status.
AUTHORS’ RESPONSE
We are grateful reviewer for this remark.
I have a few minor point questions, so please answer them.
- Is clinical stage in table 2 a prechemo stage? Is it postchemo stage? If it is a prechemo stage, why was the neoadjuvant chemo performed in stage 1?
- P6/13 spelling check –epirubicyne -> epirubicin
- Clarfy a definition of pCR adopted in this study:
No invasive tumor (+/- residual DCIS, ypT0-is) vs No carcinoma including DCIS (ypT0 only)
AUTHORS’ RESPONSE
- Clinical stage in table 2 is prechemo stage.
- I changed Epirubicine into Epirubicin.
- Complete pathological response means the lack of cancer including DCIS in specimen- ypT0N0.
The spelling was checked again as suggested.